# Cyclooxygenase-2 Blockade Is Crucial to Restore Natural Killer Cell Activity before Anti-CTLA-4 Therapy against High-Grade Serous Ovarian Cancer

**DOI:** 10.3390/cancers16010080

**Published:** 2023-12-22

**Authors:** Fernán Gómez-Valenzuela, Ignacio Wichmann, Felipe Suárez, Sumie Kato, Enrique Ossandón, Marcela Hermoso, Elmer A. Fernández, Mauricio A. Cuello

**Affiliations:** 1Department of Gynecology, School of Medicine, Pontificia Universidad Católica de Chile, Santiago 8330024, Chile; felipesuarezn@gmail.com (F.S.); skatoc@gmail.com (S.K.); eossandon@uc.cl (E.O.); 2Department of Obstetrics, School of Medicine, Pontificia Universidad Católica de Chile, Santiago 833150, Chile; wichmann@stanford.edu; 3Advanced Center for Chronic Diseases (ACCDiS), Pontificia Universidad Católica de Chile, Santiago 833150, Chile; 4Division of Oncology, Department of Medicine, School of Medicine, Stanford University, Stanford, CA 94305, USA; 5Innate Immunity Laboratory, Immunology Program, Biomedical Sciences Institute, Faculty of Medicine, Universidad de Chile, Santiago 8900085, Chile; marcehr@gmail.com; 6Fundación para el Progreso de la Medicina (CONICET), Córdoba X5000, Argentina; elmer.fernandez@unc.edu.ar; 7Facultad de Ciencias Exactas, Físicas y Naturales, Universidad Nacional de Córdoba, Córdoba X5000, Argentina; 8Center for Cancer Prevention and Control (CECAN), Santiago 8330023, Chile

**Keywords:** high-grade serious ovarian cancer, tumor immune microenvironment, cyclooxygenase-2, cytotoxic T-lymphocyte-associated protein-4, immunotherapy, NK cells

## Abstract

**Simple Summary:**

Cyclooxygenase-2 (COX-2) expression causes several changes in the tumor microenvironment (TME) of ovarian cancer, which are associated with a low immunotherapy response in patients and immunosuppression in the TME, mainly due to cytotoxic T-lymphocyte-associated-protein-4 (CTLA-4) expression. However, its effect on the immune component of the TME has not been fully elucidated. In this study, using several integrated bioinformatic tools, we analyzed public transcriptomic data from four groups of ovarian cancer patients. We found that high expression of COX-2 is linked to poor survival, changes in the immune ecosystem, cell dysfunction, and lower effector activity of natural killer (NK) cells. Afterwards, we validated these results using flow cytometry and cytotoxicity assays on circulating NK cells from HGSOC patients. These cells were co-cultured with HGSOC cell lines while undergoing COX-2 and CTLA-4 blockade. Our results suggest that targeting COX-2 prior to the anti-CTLA-4 immunotherapy scheme, which takes advantage of NK cells cytotoxic capacity, may be a promising strategy to improve the effectiveness of immunotherapy for ovarian cancer patients.

**Abstract:**

Chronic inflammation influences the tumor immune microenvironment (TIME) in high-grade serous ovarian cancer (HGSOC). Specifically, cyclooxygenase-2 (COX-2) overexpression promotes cytotoxic T-lymphocyte-associated protein-4 (CTLA-4) expression. Notably, elevated COX-2 levels in the TIME have been associated with reduced response to anti-CTLA-4 immunotherapy. However, the precise impact of COX-2, encoded by *PTGS2*, on the immune profile remains unknown. To address this, we performed an integrated bioinformatics analysis using data from the HGSOC cohorts (TCGA-OV, *n* = 368; Australian cohort AOCS, *n* = 80; GSE26193, *n* = 62; and GSE30161, *n* = 45). Employing Gene Set Variation Analysis (GSVA), MIXTURE and Ecotyper cell deconvolution algorithms, we concluded that COX-2 was linked to immune cell ecosystems associated with shorter survival, cell dysfunction and lower NK cell effector cytotoxicity capacity. Next, we validated these results by characterizing circulating NK cells from HGSOC patients through flow cytometry and cytotoxic assays while undergoing COX-2 and CTLA-4 blockade. The blockade of COX-2 improved the cytotoxic capacity of NK cells against HGSOC cell lines. Our findings underscore the relevance of COX-2 in shaping the TIME and suggest its potential as a prognostic indicator and therapeutic target. Increased COX-2 expression may hamper the effectivity of immunotherapies that require NK cell effector function. These results provide a foundation for experimental validation and clinical trials investigating combined therapies targeting COX-2 and CTLA-4 in HGSOC.

## 1. Introduction

High-grade serous ovarian cancer (HGSOC) is the deadliest gynecologic malignancy, with an overall 5-year survival rate below 40% in most countries, despite the recent incorporation of poly (ADP-ribose) polymerase inhibitors (PARPi) and immune checkpoint blockers (ICB) [1]. This challenge has led to considerable efforts aimed at deepening our understanding of the interactions between immune cells and other cellular components of the tumor immune microenvironment (TIME), including stromal cells, which critically influence the antitumor immune response in this context [2].

A higher abundance of tumor-infiltrating lymphocytes (TILs) is associated with a better prognosis in HGSOC [3]. However, advanced stages of HGSOC often exhibit systemic inflammation that nullifies the predictive value of TILs [3] and fosters an immunosuppressive immune infiltration pattern characterized by regulatory T lymphocytes (Tregs) [4,5]. Growing evidence shows that the constitutive membrane expression of cytotoxic T-lymphocyte-associated protein-4 (CTLA-4) by Tregs [6] promotes immune escape by competing with the co-stimulatory molecule CD28 present on conventional CD4+ and CD8+ T cells for the CD80/86 ligands, preventing T cell activation [6,7,8,9].

CTLA-4 blockade has emerged as a promising immunotherapeutic approach in cancer treatment. Ipilimumab, a monoclonal antibody targeting CTLA-4, has shown significant clinical efficacy by enhancing T-cell activation and unleashing antitumor immune responses [10]. While CTLA-4 blockade disrupts the suppressive activity of Tregs, recent studies suggest broader effects on other cell populations, including CD4+ T lymphocytes that produce effector cytokines, such as interleukin (IL)-2, promoting natural killer (NK) cell activity [11,12,13].

The blockade of CTLA-4 has been successfully implemented in various solid tumors, such as melanoma and hepatocellular carcinoma [14]. However, although the expression of CTLA-4 in HGSOC is similar to other solid tumors [15], numerous clinical trials have evaluated anti-CTLA-4 treatment schemes in HGSOC without demonstrating advantages over different therapeutic approaches [16]. Therefore, understanding compensatory signals and dysfunctional cellular pathways influencing anti-CTLA-4 response in HGSOC patients is crucial.

The cyclooxygenase-2 (COX-2)/prostaglandin E2 (PGE2) axis is a crucial modulator of low-grade chronic inflammation, orchestrating the inflammatory characteristics of the TIME and contributing to the development and progression of diverse cancers [17,18,19,20,21]. The COX-2/PGE2 axis is an inducible inflammatory signaling cascade in response to numerous inflammatory mediators (i.e., tumor necrosis factor-alpha (TNF-α), IL-6, among others), oncogenes, and growth factors [18,22]. Multiple studies indicated significant activity of the COX-2/PGE2 axis in the TIME of HGSOC, the expression of which promotes the malignant behavior of ovarian cancer cell lines [23,24]. In addition, activation of the COX-2/PGE2 axis promotes invasion, proliferation, and epithelial-to-mesenchymal transition (EMT) in HGSOC cells [24]. Moreover, in pan-Cancer studies, elevated COX-2 expression is an independent predictor of survival and response to ICB [25,26,27].

Evidence supports that this therapeutic regimen stimulates antitumor inflammatory response pathways [26]. Furthermore, it shifts the infiltrating inflammatory cells towards preferentially effector phenotypes [27].

To address the knowledge gap regarding the specific changes in the immune profile caused by COX-2 in the TIME of HGSOC, we conducted an integrated bioinformatics analysis using clinical and genomic data from four HGSOC cohorts. Our study aims to investigate whether the expression of COX-2, encoded by the gene *PTGS2*, influences the immune profile and hampers natural killer (NK) cell activity, potentially affecting the efficacy of anti-CTLA-4 immunotherapy in HGSOC patients. We utilized Gene Set Variation Analysis (GSVA), MIXTURE and Ecotyper cell deconvolution algorithms to evaluate the effects of *PTGS2* expression on the immune profile, quantifying the association between *PTGS2* and *CTLA4* as a ratio and correlating it with cell type abundance, gene expression signature scores, and different ecosystems. Lastly, to validate bioinformatics results, we isolated circulating NK cells from *n* = 5 HGSOC patients and assessed the effect of COX-2 and CTLA-4 blockade over phenotype and cytotoxic capacity against HGSOC cell lines in an in vitro co-culture model.

## 2. Materials and Methods

### 2.1. Data Collection

We collected RNA-seq expression data (transcripts per million, TPM) and corresponding clinical information from the Genomic Data Commons (GDC) using the TCGAbiolinks R package [28]. Additionally, we retrieved gene expression RNA-seq data (Fragments Per Kilobase of transcript per Million mapped reads Upper Quartile, FPKM-UQ) from AOCS samples from the ICGC Data Portal. We transformed to TPM using the ‘fpkmToTpm_matrix’ function from the GeoTcgaData R package [29]. To complement the analysis, we obtained two quantile-normalized public ovarian cancer microarray datasets, GSE30161 and GSE26193, from GEOquery in Bioconductor for downstream survival analyses. All datasets were filtered to include only the ovary as the primary tumor site, a histology typification as “high-grade serous ovarian cancer”, and stages III and IV with clear and complete clinical information.

### 2.2. Genetic Assessment of COX-2, CTLA-4, and Ovarian Tumor Immune Microenvironment

To build up the survival curves (using progression-free survival [PFS] as the primary clinical outcome), we used the KM plotter Survival Analysis Tool [30], considering the ratio of the prostaglandin-endoperoxide synthase 2, *PTGS2* (the COX-2 gene) and *CTLA4*, and defined survival according to the median-based *PTGS2*/*CTLA4* cutoff.

Cell deconvolution was performed using MIXTURE, a noise-constrained recursive feature selection algorithm based on ν-support vector regression [31]. The LM22 signature matrix [32] was utilized to estimate the relative abundance of 22 distinct immune cell types within the TIME, providing cell proportions. This deconvolution tool presents comparative advantages in assessing immune cell types in RNA-Seq databases [33]. Additionally, we conducted Gene Set Variation Analysis (GSVA) [34], (i) to analyze immune cell signature scores in RNA array datasets, leveraging existing pan-Cancer single-cell immune cell Atlas signatures [35], and (ii) to evaluate the enrichment of the *PTGS2* pathway. We also examined cell states and their interactions in TCGA and Gene Expression Omnibus (GEO) ovarian cancer datasets using Ecotyper (Stanford University, Stanford, CA, USA) [36].

### 2.3. Patients

Human fresh whole blood from *n* = 5 HGSOC patients was collected based on the Institutional Review Board of Pontificia Universidad Católica de Chile protocol (ID 190408002, approved 05/07/2020) for studies involving humans. We isolated fresh peripheral blood mononuclear cells (PBMC) through Ficoll gradient in 50 mL SepMate Isolation Tubes (86450, Stemcell Technologies, Vancouver, BC, Canada).

### 2.4. NK Cells Isolation and Cell Cultures

NK cell isolation was performed using the EasySep Isolation Kit (17955, Stemcell Technologies, Vancouver, BC, Canada) from PBMC following the manufacturer’s recommendations. Isolated NK cells were cultured in 24-well treated clear plates (3526, Costar Corning, NY, USA), adding 800 μL of Roswell Park Memorial Institute (RPMI) 1640 without Fetal Bovine Serum (FBS) supplemented with 1× penicillin/streptomycin (pen/strep) plus 100 μL of autologous human serum, and 50 μL or 500 IU of IL-2 (1 mg/mL). We cultured human cancer cell lines HeyA8 kindly donated by Dr. Gloria Huang from the Albert Einstein College of Medicine, Bronx, NY, USA (harboring TP53 P72R polymorphism and authenticated using the Genemarker 10 kit [Promega], RRID: CVCL_8878), and SKOV3 (American Type Culture Collection [ATCC], Manassas, VA, USA) in RPMI 1640 with 10% of FBS supplemented with 1× pen/strep and 1× GlutaMAX (35050061, Gibco BRL, Life Technologies AG, Basel, Switzerland).

### 2.5. Drugs Assessment

We performed the drug assessment using selective COX-2 inhibitor celecoxib (SML3031, Merck, Darmstadt, Germany) at a concentration of 5 mM and the anti-CTLA-4 monoclonal antibody ipilimumab (A2001, Selleckchem, Houston, TX, USA) at a concentration of 80 nM. These dose concentrations of celecoxib and ipilimumab represent values usually achieved clinically.

### 2.6. Treatment Protocol

After three days of incubation, isolated NK cell culture was stimulated with three pulses of celecoxib (CXB) or MOCK daily. Ipilimumab or MOCK was added at the third pulse. The HGSOC cell lines, SKOV3 and HeyA8, were treated similarly. We differentiated four experimental strategies as follows: 1, Control; 2, CXB; 3, Ipilimumab; 4, CXB plus ipilimumab.

### 2.7. Immunophenotyping of NK Cells

Flow cytometric analysis was performed using the BD FACSCanto™ II Flow Cytometry System cytometer and analyzed using FlowJo (v.X 10.0.7r2, Tree Star, San Carlos, CA, USA) software. The following antibodies were used to determine the immunophenotype of NK cells from HGSOC patients. For cell surface staining we used: CD3 PerCP (clone OKT3, BioLegend, San Diego, CA, USA); CD56 PE-Cy7 (clone 5.1H11, BioLegend, San Diego, CA, USA); CD16 FITC (clone 3G8, BioLegend); CTLA-4 BV421 (clone BNI3, BioLegend); and CD107a PerCP-Cy5.5 (clone H4A3, BioLegend). For intracellular staining, we used Fixation/Permeabilization Kit (BD Cytofix/Cytoperm, San Jose, CA, USA) following manufacturer’s indications. Then, we stained for COX-2 PE (clone D5H5, Cell Signaling, Denvers, MA, USA). For the cell viability assay, we used Zombie NIRFixable Viability Kit at a 1:750 dilution (423117, BioLegend) following manufacturer’s indications.

### 2.8. Cytotoxicity Assay in NK Cells and HGSOC Cell Lines Co-Culture

After 24 h of the third CXB pulse, NK cells and HGSOC cell lines were co-cultured in a 1:10 proportion for four hours. HGSOC cytotoxicity was assessed by the release of the soluble cytosolic enzyme lactate dehydrogenase (LDH), using the LDH-Glo Cytotoxicity Assay (J2380, Promega, Madison, WI, USA), following the manufacturer’s indications.

### 2.9. Statistical Analyses

We performed all data analyses using GraphPad Prism (v.9.3.0, Dotmatics, Boston, MA, USA) and R software (version 4.3.0). Group comparisons for continuous variables were conducted using Welch’s *t*-test and Kruskal Wallis’ test (after analyzing variance through Bartlett’s test). Specifically, for cytotoxicity assays and to determine the variance according to each experimental group, we conducted a two-way ANOVA test with Tukey’s multiple comparisons. In addition, Chi-squared tests were used for categorical variable comparisons using the ggstatsplot R package [37]. Univariate/multivariate Cox proportional hazard regression analyses were performed for all variables using the ‘coxph’ function from the survival R package version 3.5–5. A *p*-value ≤ 0.05 was considered statistically significant.

## 3. Results

### 3.1. The PTGS2/CTLA4 Ratio Defines Survival in Multiple HGSOC Cohorts

We first examined the expression of *PTGS2* and *CTLA4* in normal and tumor tissue samples for diverse types of cancer, employing the TNMplot tool [38] of the KMplotter web platform [30]. We noted that the expression levels of *PTGS2* and *CTLA4* were highly heterogeneous across various cancer types. However, we observed that the expression pattern of both genes was consistent in tumor tissues of gynecological cancers (i.e., Uterine Carcinosarcoma [UCS] and Uterine Corpus Endometrial Carcinoma [UCEC]) compared with tumor-adjacent controls, and this difference was more noticeable for *CTLA4* (Figure 1A). Interestingly, although we did not observe an increase in *PTGS2* expression in tumor tissue, there was a weak positive correlation between *CTLA4* in normal tissues and HGSOC (Figure 1B). In a scenario where patients with HGSOC are candidates for anti-CTLA-4 immunotherapy due to high CTLA-4 expression, and presuming that *PTGS2* and *CTLA4* exhibit a ‘non-linear association’, the *PTGS2/CTLA4* ratio allows us to directly investigate how *PTGS2* influences changes in the immune profile of these patients, and whether this would prevent an optimal response to anti-CTLA4 therapy. *PTGS2* and *CTLA4* did not define survival in the four cohorts analyzed (Appendix A).

Assuming high levels of *CTLA4* presented in these patients, the ratio will only be affected by changes in the *PTGS2* level. Additionally, this approach allows us to identify the effects of chronic inflammation, represented by *PTGS2* levels, on the acute or lymphocytic tumor infiltrate, which is represented by *CTLA4*.

Utilizing the *PTGS2*/*CTLA4* ratio, we stratified samples into high- and low-expression groups based on the median ratio to evaluate the impact on survival in multiple ovarian cancer cohorts (Figure 2).

### 3.2. A High PTGS2/CTLA4 Ratio Determines the Abundance and Cell States of Immune Cells in HGSOC

To search for potential mechanisms associated with reduced survival in patients with a high *PTSG2*/*CTLA4* ratio, we evaluated the relationship between this ratio and changes to the cell neighborhoods and functionality of the antitumor immune response in the same datasets. To this end, we used Ecotyper as a proxy to identify different cell states (or S) and cellular communities (cancer ecotypes or CE). Briefly, this tool performs high-level analysis of cellular composition and interactions from bulk transcriptomics data, enabling the identification of functional cell states and CEs related to survival outcomes and therapy response in multiple cancer types [36]. Interestingly, Luca et al. developed a pan-cancer analysis in TCGA to identify carcinoma ecotypes based on RNA expression, to represent better or worse prognoses [36]. Thus, for ovarian cancer, Ecotyper defined certain carcinoma ecotypes associated with a better prognosis or response to treatment (CE09 and CE10) and, in contrast, other carcinoma ecotypes associated with lower survival or poor response to treatment (CE03 and CE06). The rest of the ecotypes would not significantly influence the prognosis or the response to treatment in ovarian cancer.

The *PTGS2*/*CTLA4*-high group from the TCGA dataset exhibited a trend for a more significant proportion of CE06 ecotype, characterized by high fibroblast abundance, smoking-related mutations, lower survival, and reduced therapy response (Figure 3A). In contrast, this group showed a lower proportion of CE09/CE10 ecotypes than *PTGS2*/*CTLA4*-low patients (Figure 3A). These CEs are associated with better survival outcomes and therapy response due to their proinflammatory profile (leukocyte enrichment) and higher immunoreactivity. Similar trends were observed in other cohorts, although not statistically significant due to smaller sample sizes (Figure 3B–D and Table 1).

Hence, we analyzed the immune cells most relevant to survival in ovarian cancer. Subsequently, we categorized these immune cells based on *PTGS2*/*CTLA4* ratios to investigate their potential impact on outcomes. According to Ecotyper, some cell states are more associated with survival. For ovarian cancer, Ecotyper defined “state S01” as functional NK cells, related to better survival. In contrast, S03 state is associated with worse survival or response to treatment. The rest of the NK cells do not significantly affect these variables.

Notably, the S01 NK cell state was significantly lower in the *PTGS2*/*CTLA4*-high group in the TCGA ovarian cancer dataset (Figure 4A). However, this trend was not observed in the other cohorts (Figure 4B–D and Table 2). To highlight, we did not observe differences in CD8+, monocyte/macrophage, dendritic cells, or plasma cells (Appendix A).

Given that the S01 NK cell state is part of the CE09 ecotype, we measured differences in immune cell type abundance between *PTGS2*/*CTLA4* ratio groups in HGSOC. We generally observed plenty of immune cells in the *PTGS2*/*CTLA4*-low group (Appendix A). Nevertheless, these phenomena changed when we studied NK cells. We found that the total count of activated NK cells (estimated by MIXTURE) was higher only in the *PTGS2*/*CTLA4*-low group from the ovarian cancer TCGA cohort (Figure 5A), but not for the AOCS cohort (Figure 5B). Based on the above, we wondered if there was any element related to the expression of COX-2 that could explain this phenomenon. Hence, we analyzed the COX-2 pathway using GSVA (i.e., *PTGER2*, *PTGER4*, *PTGIS*, and *PTGES* genes), and we noted that the AOCS cohort was the only dataset that displayed a significant enrichment of the COX-2 pathway when *PTGS2*/*CTLA4* ratio was high (Appendix A).

Last, we contrasted these results with the GSVA analysis for NK cell estimation based on Nieto’s signature [35], which describes the development of a single-cell tumor immune atlas, a comprehensive map of the TIME in 13 different cancer types. The atlas was generated by combining single-cell RNA sequencing and spatial transcriptomics data. It provided a wealth of information about the composition and function of the TIME in different cancer types. The observation was consistent with NK cell GSVA scores (Figure 5C,D). We, therefore, concluded that *PTGS2*/*CTLA4* does not affect the estimated NK cells in HGSOC.

### 3.3. A High PTGS2/CTLA4 Ratio Hinders NK Gene Score Associated with NK Infiltration, Cytotoxicity, and Survival in HGSOC

To assess the impact of the *PTGS2*/*CTLA4* ratio on NK cell infiltration and survival in HGSOC, we utilized a gene signature reported by Cursons et al. [39]. This signature represents several genes associated with NK cell chemotaxis, cytotoxicity, and tissue infiltration (Figure 6A). To date, it has not been evaluated in HGSOC. As a result, all cohorts analyzed displayed a lower NK score enrichment when the *PTGS2*/*CTLA4* ratio was elevated (Figure 6B–E). Next, we examined the relationship between these genes and *PTGS2* to define survival in HGSOC. *PTGS2* determined a shorter survival when it was inversely expressed to the NK score (i.e., *PTGS2* expression increases as NK score decreases) (Figure 6F).

Considering the above, we analyzed whether the NK score could modulate crucial clinical variables in the survival of patients with HGSOC. For this, we evaluated clinical data from the TCGA ovarian cancer cohort, such as age (over or under 70 years), FIGO stage (III or IV), debulking (optimal or sub-optimal), and response to treatment (responds or does not respond). Through different multivariable models, we discovered that the NK score determines survival, so a low NK score further decreases the survival of non-responders and sub-optimal debulking patients (Figure 7A,B). These results were consistent with other statistical models analyzed (Figure 7C–E).

### 3.4. Selective COX-2 Blockade through Celecoxib Improves NK Cells’ Cytotoxic Capacity Independent of CTLA-4 Signaling

As an initial step, we isolated circulating NK cells from *n* = 20 HGSOC patients and *n* = 3 healthy donors to assess the basal expression of COX-2, CTLA-4, and CD107a using flow cytometry. Despite the differing sample sizes between the experimental groups, our observations indicated that CTLA-4 expression was present in a limited subset of HGSOC patients (7 out of 20) (Appendix A).

Next, from new *n* = 5 HGSOC patients whose circulating NK cells demonstrated CTLA-4 expression, we treated the NK cells with CXB, ipilimumab, and their combination, according to the experimental scheme described in Methods (briefly, three pulses of CXB separated by 24 h, where the third pulse of CXB was accompanied with ipilimumab).

The expression level of NK markers measured by Median Fluorescence Intensity (MFI) did not present differences between the experimental groups. Nevertheless, as seen in Figure 8A, when we analyzed the percentage of positive cells for each marker according to the experimental groups, we noticed that CXB promoted trending increases in the proportion of cells positive for the cytotoxic capacity marker CD107a compared to the other experimental groups. Moreover, when we specifically analyzed NK cells that express COX-2, we found that the two major NK cell populations based on their cytotoxic capacities, that is CD56^dim^CD16^+^ (high capacity) and CD56^bright^CD16^+^ (low capacity), presented this trend due to an increase in the expression of CD107a after CXB stimulation (Figure 8B,C).

Last, to validate both our in-silico experiments and the flow cytometry results, we evaluated the activity of circulating NK cells derived from two new HGSOC patients using functional cytotoxicity assays. For this, we performed a co-culture of NK cells from two new HGSOC patients UC251 and UC255 with HeyA8 and SKOV3 cell lines, respectively. Each independent culture received CXB and ipilimumab treatments described in the methodology, to later be co-cultivated for 4 h. Next, we evaluated cytotoxicity on *n* = 3 replicates for each condition. Moreover, we determined which condition describes a greater variance in our adjusted model for cytotoxicity assay, adjusting to the HeyA8 and SKOV3 cell lines and NK cells experimental group for each patient through a two-way ANOVA test with Tukey’s multiple comparisons.

Interestingly, previous treatment with CXB on NK cells corresponded to the variable that mainly explained the difference in cytotoxicity of both cell lines. For HeyA8, the contrast between the “CXB vs. Control” conditions in the context of the NK cells variable was statistically significant (*p*-value= 0.0253) (Figure 9A). Moreover, for the SKOV3 cell line, CXB treatment was borderline significant compared to its control (*p*-value= 0.0524) (Figure 9B). Last, we noted that ipilimumab or combined treatment did not promote a greater cytotoxicity of HGSOC cell lines.

## 4. Discussion

Our study provides compelling evidence that COX-2 plays a significant role in shaping the immune profile and NK cell activity in HGSOC, which has potential implications for immunotherapy responses and patient outcomes. These findings underscore the importance of considering the COX-2/PGE2 axis as a promising therapeutic target in HGSOC, and shed light on the intricate immune mechanisms associated with the *PTGS2*/*CTLA4* ratio.

Several studies have shown that the COX-2/PGE2 axis reduces the cytotoxic capacity and the production and response to inflammatory cytokines in NK cells [40,41,42]. In this fashion, Martinet et al. showed that PGE2 binds to EP2 and EP4 receptors, inhibiting natural cytotoxicity receptors of NK cells, including NKp30, NKp46, CD16, NKG2D, and CD107a, and confirming the hampering of their cytotoxic activity by cyclic adenosine monophosphate (cAMP)-dependent activation of protein kinase A (PKA) type I in murine models [40]. Hence, increased COX-2/PGE2 signaling may alter antitumor effector machinery even in immune-enriched TIME. Furthermore, activation of the EP4 receptor by PGE2-induced *FOXP3* gene expression of Tregs in mouse models and, more importantly, COX-2 blockade decreases the proportion of Tregs in TILs [43]. In conclusion, the inflammatory environment, both local and systemic, may modulate the response to immunotherapeutic schemes, and it might even limit T cell-mediated immunity after cytotoxic treatment [27].

Interestingly, we noted that *CTLA4* gene expression represents infiltration and not necessarily immunosuppression in this analysis, mainly because the low *PTGS2*/*CTLA4* group presented a higher estimation of immune cells. This conclusion allows us to infer that COX-2 overexpression would affect the effectiveness of therapeutic options that stimulate the cellular adaptive immune response. For example, the anti-CTLA-4 antibody ipilimumab has opened an alternative therapeutic opportunity since its ability to induce antibody-dependent cellular cytotoxicity (ADCC) has been described [44,45,46]. Briefly, NK cells recognize the Fc fragment of ipilimumab through the Fc gamma receptor IIIa (FcγRIIIa) or CD16 receptor, which promotes its cytotoxic effector activity [47]. Consequently, combining COX-2/PGE2 blockade with CTLA-4 blockade would provide a comparative advantage over anti-PD-1/anti-PD-L1 strategies, principally due to anti-CTLA-4 exhibiting extended durability of the antitumor response [48,49]. In addition, CTLA-4 blockade may favor the expansion of selective T cell receptor (TCR) clones in lymphocytes [12], which would be advantageous for modeling targeted adaptive antitumor immune responses. In addition, current evidence suggests that the combined use of ICB immunotherapy and PARPi could improve immunogenicity in ovarian cancer [50]. However, ADCC inherently requires the proper functionality of effector cells, particularly NK cells that recognize CTLA-4 through its CD16 receptor. In addition, improving the effector capabilities of NK cells may impact other immunotherapies currently evaluated in HGSOC and other gynecologic cancers. Thus, future research could explore the use of anti-CTLA-4 and PARPi therapies in combination with selective blockers of the COX-2/PGE2 axis.

To date, the precise effects of anti-CTLA-4 monoclonal antibodies on human NK cells are unknown; however, some works suggest that anti-CTLA-4 would mainly promote an immunomodulatory profile associated with the release of proinflammatory cytokines such as interferon-gamma (IFN-γ) and TNF-α, over changes in the promotion of the cytotoxic capacity of NK cells [51,52]. Furthermore, it is suggested that NK cells may be involved in the therapeutic success of anti-CTLA-4 antibodies through indirect mechanisms, specifically since transitory cell-membrane CTLA-4 expression makes it difficult to characterize NK cells correctly [53].

Regarding the bioinformatics tools used, the MIXTURE algorithm allowed us to discover differences in the estimation of NK cells in the RNA-seq datasets of TCGA ovarian cancer patients. Surprisingly, the trend of a higher estimate of immune cells in a low *PTGS2*/*CTLA4* ratio is missing for NK cells. In addition, one of our work’s intriguing findings corresponds to the enrichment of the *PTGS2* pathway in the AOCS cohort. In this sense, we assume that other elements associated with the *PTGS2* pathway can influence the inflammatory response, mainly promoting the recruitment of immune cells within the TME of HGSOC. Nevertheless, despite no quantitative differences in cell amounts, a high *PTGS2*/*CTLA4* ratio was associated with a low NK signature score in the AOCS cohort.

These results, complemented with the functional analyses in Ecotyper, indicate that an increase in the expression of COX-2 may hamper the cytotoxic capacities of NK cells in HGSOC. Hence, we wanted to evaluate if the *PTGS2*/*CTLA4* ratio also determines the infiltration and cytotoxic capacity estimate in HGSOC. For this, we considered the recent publication by Zhang et al. [54]. In that work, the authors proposed several genes associated with risk prognosis models through differential expression analysis. Interestingly, their results involved genes not necessarily associated with NK cell activity. Despite being a remarkable and first advance to estimate NK cell functionality in HGSOC, GSVA analysis using this score was only robust for RNA-Seq-based databases. Likewise, we incorporated the NK score described by Cursons et al. [39], whose results were cross-sectional in all the cohorts analyzed. We included this score in predictive risk models and useful clinical variables based on the above. We observed that the COX-2/PGE2 axis overexpression was associated with reduced NK score and dysfunctional NK cells within the TIME of HGSOC. Given the above, recognizing the expression levels of COX-2 in ovarian tissue could be a valuable practice in the development of personalized therapeutic approaches, complementing the current biological diagnostic methods that rely on analyzing tumor expression of cancer antigen 125 (CA 125) and Human Epididymis Protein 4 (HE4) [55]. Moreover, incorporating the assessment of each patient’s inflammatory state could provide valuable insights for screening and treating patients affected by ovarian cancer.

The main limitations of this study are the cross-sectional analytical design of a small number of cohorts and the inability to represent tumor-infiltrating immune cells due to the LM22 signature developed from peripheral blood mononuclear cells (PBMCs). Nevertheless, various efforts are being made to create new datasets of public signature matrices using downsized public single-cell RNAseq datasets in pan-Cancer analysis [56,57,58]. In addition, we recognize the significance of augmenting the patient sample size to ensure a more comprehensive and valid study, consequently strengthening the importance and translational relevance of our proposal. Among the strengths, we integrated various bioinformatics tools to analyze widely legitimated databases. In addition, through these tools, we examined a little-explored biological process that would affect the prognosis of patients with HGSOC.

Last, we must also emphasize that the modulation of COX-2 expression, as a representation of chronic inflammation, may require a prolonged therapeutic regimen, which could promote synergism with anti-CTLA-4 to improve NK cell activity against HGSOC tumor cells.

## 5. Conclusions

This work represents the first evidence of the influence of COX-2 on NK cell function as a crucial element in the efficacy of anti-CTLA-4 therapy for ovarian cancer patients that exhibited high tumoral levels of CTLA-4. Therefore, our in silico findings and in vitro validation support the rationale for conducting preclinical studies.

Based on our findings, we believe the modulation of chronic inflammation to direct an immune pattern with greater antitumor capacity in HGSOC is relevant. In summary, we advocate for targeted and personalized immune characterization of patients with HGSOC before initiating anti-CTLA-4 immunotherapy to establish an effective therapeutic strategy.

This intervention may not only improve the response to treatments but also improve the prognosis of ovarian cancer patients.

## Figures and Tables

**Figure 1 cancers-16-00080-f001:**
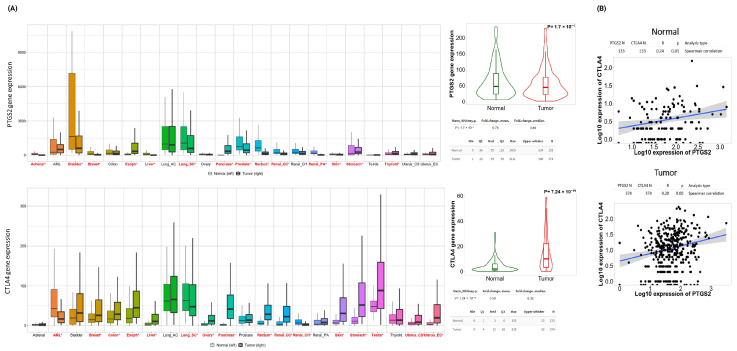
Pan-cancer comparison of *PTGS2* and *CTLA4* expression in available normal and tumor samples: (**A**) RNA-Seq data, and (**B**) Spearman correlation analysis of *PTGS2* and *CTLA4* in normal ovarian tissue and ovarian tumor. Analysis realized in TNMplot web tool created by Bartha et al. [38]. Red letters and * symbol indicate a statistical difference between tumor and normal tissue.

**Figure 2 cancers-16-00080-f002:**
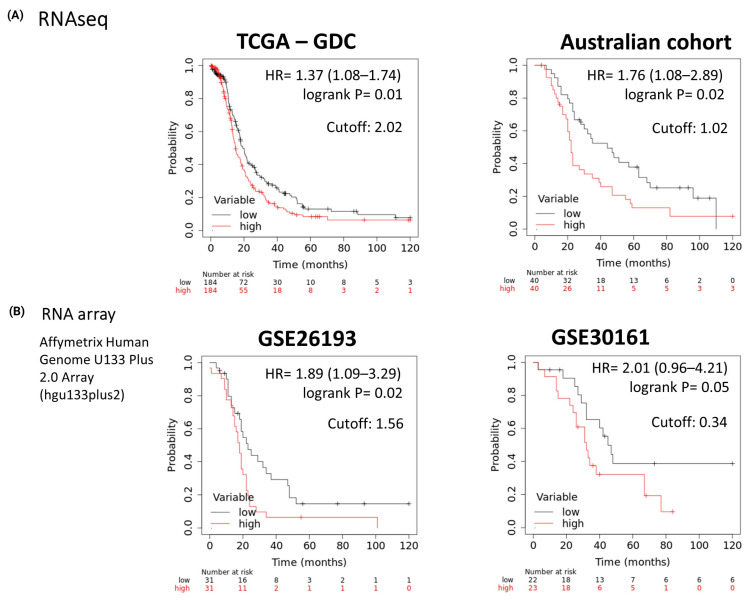
Kaplan-Meier plot defined by Progression Free Survival (PFS) of ovarian cancer patients based on (**A**) RNA-Seq (TCGA-GDC, Australian cohort) and (**B**) RNA arrays (GSE26193, GSE30161) datasets according to median expression (defined cutoff) of *PTGS2*/*CTLA4* ratio.

**Figure 3 cancers-16-00080-f003:**
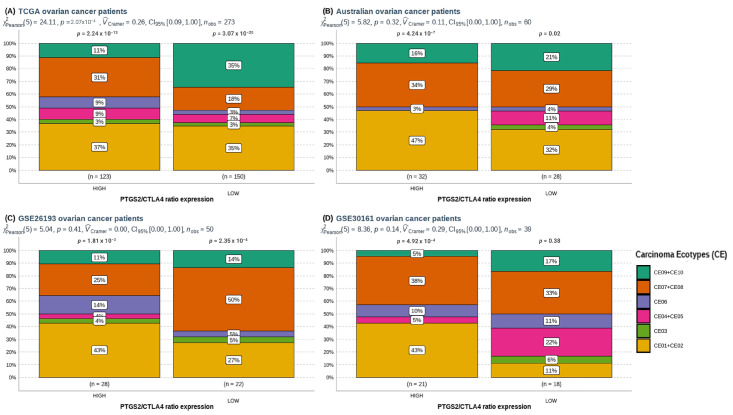
Prevalence comparison of carcinoma ecotypes (CE) in (**A**) TCGA-GDC, (**B**) Australian patients, (**C**) GSE26193, (**D**) GSE30161 ovarian patients’ cohorts according to high or low *PTGS2*/*CTLA4* ratio. For comparison, we grouped CE associated with good prognosis/response to treatment (CE09 + CE10), and analyzed separately those who are associated with worse prognosis/response to treatment (CE03 and CE06). Pearson’s chi-squared test was performed through ggstatsplot R package.

**Figure 4 cancers-16-00080-f004:**
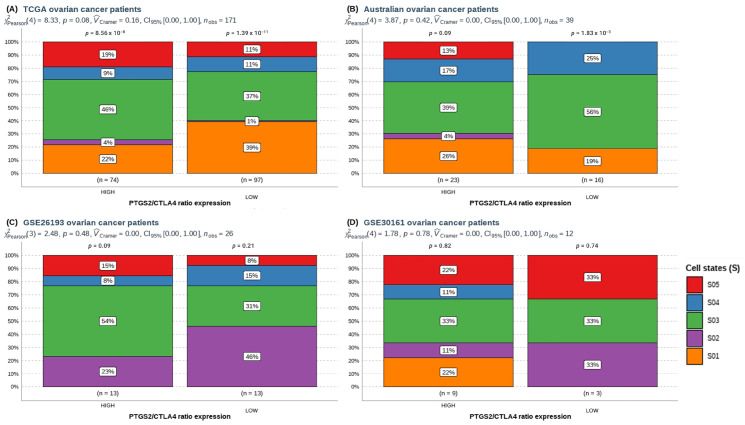
Prevalence comparison of NK cell states (S) defined in patients from (**A**) TCGA-GDC, (**B**) AOCS patients, (**C**) GSE26193, (**D**) GSE30161 ovarian patients’ cohorts according to high or low *PTGS2*/*CTLA4* ratio expression. Pearson’s chi-squared test performed through ggstatsplot R package.

**Figure 5 cancers-16-00080-f005:**
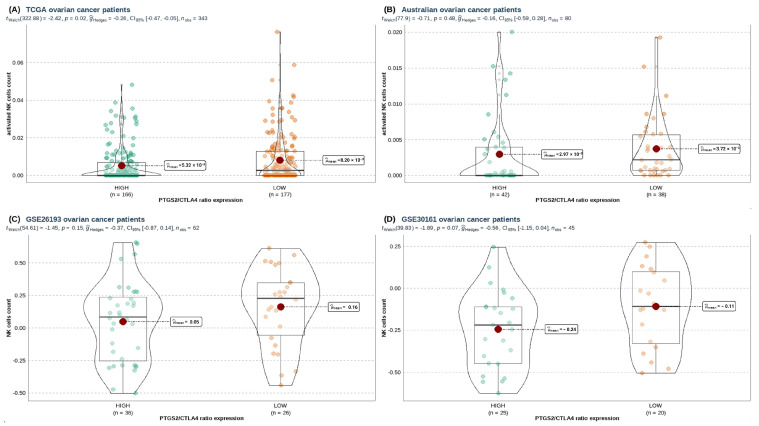
Comparison of MIXTURE’s activated NK cell estimation for (**A**) TCGA-GDC and (**B**) AOCS cohorts; (**C**) GSVA enrichment of NK cells from Nieto’ signature for GSE26193 and (**D**) GSE30161 ovarian patients’ cohorts. Selection of groups according to PTGS2/CTLA4 ratio expression. Welch’s *t*-test performed through ggstatsplot R package.

**Figure 6 cancers-16-00080-f006:**
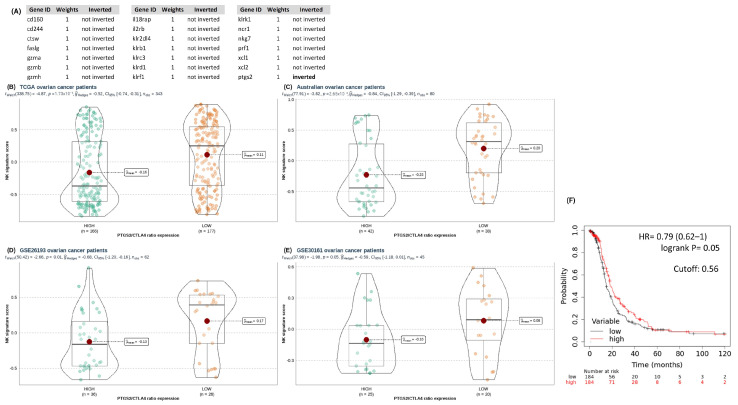
(**A**) NK score gene list for survival analysis inversely related with *PTGS2* gene. Comparison of GSVA enrichment of NK score signature according to *PTGS2*/*CTLA4* ratio for (**B**) TCGA-GDC, (**C**) AOCS cohorts, (**D**) GSE26193, and (**E**) GSE30161. (**F**) Kaplan-Meier plot defined by Progression Free Survival (PFS) of TCGA-GDC ovarian cancer patients according to median expression (defined cutoff) of NK score expression. Welch’s *t*-test performed through ggstatsplot R package.

**Figure 7 cancers-16-00080-f007:**
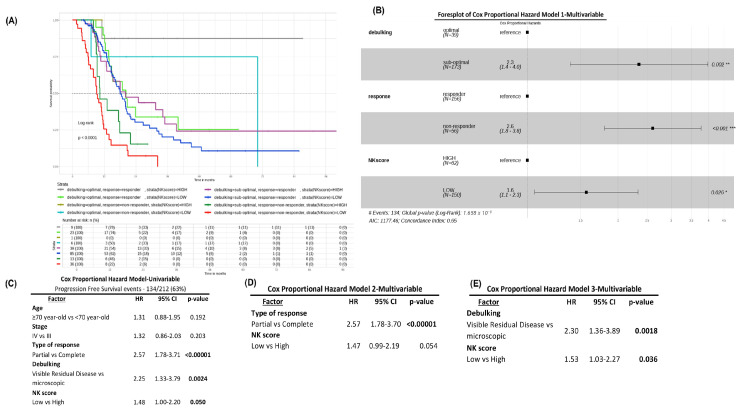
Disease Free Progression (PFS) (**A**) curve and (**B**) forest plot for multivariate Cox proportional hazard regression analyses based on debulking, response to treatment, and NK score. (**C**–**E**) Univariate and multivariate Cox proportional regression analyses carried out to assess NK score as an independent risk factor. * *p* < 0.05, ** *p* < 0.01, *** *p* < 0.001.

**Figure 8 cancers-16-00080-f008:**
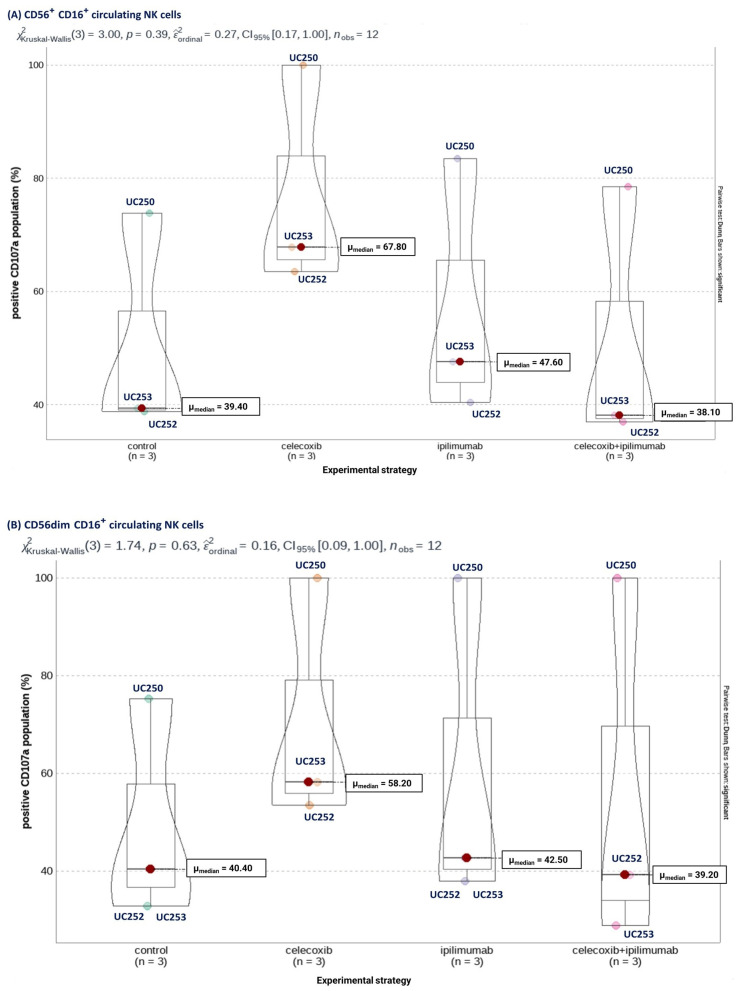
Comparison of the mean percentage of CD107a+ in (**A**) total CD56^+^CD16^+^, (**B**) CD56^dim^CD16^+^, and (**C**) CD56^bright^CD16^+^ circulating NK cells, according to experimental strategy. Kruskal-Wallis test. Significance was defined as *p*-value < 0.05.

**Figure 9 cancers-16-00080-f009:**
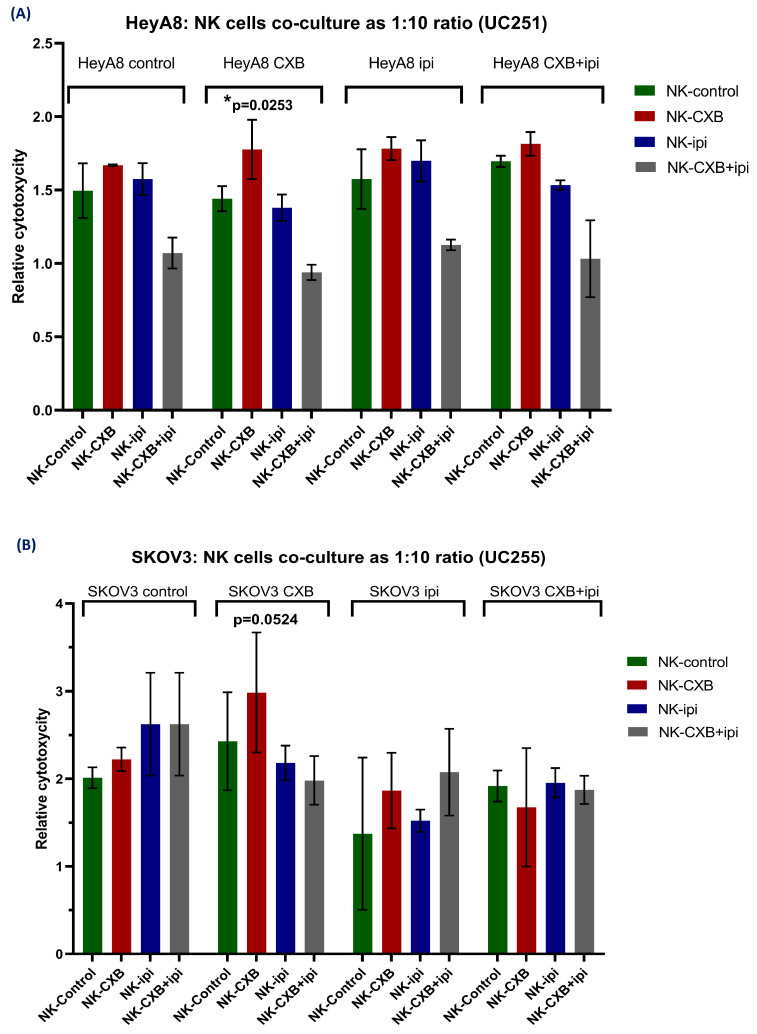
Relative cytotoxicity (compared to independent cell cultures) of (**A**) HeyA8 and (**B**) SKOV3 cell line co-cultured with circulating NK cells from HGSOC patients UC251 and UC252, respectively. The cytotoxicity was measured by bioluminescence for LDH released into the culture medium. Each bar represents *n* = 3 replicates with standard deviation. CXB, celecoxib. Ipi, ipilimumab. Two-way ANOVA with Tukey’s multiple comparisons. Significance was defined as *p*-value < 0.05.

**Table 1 cancers-16-00080-t001:** Carcinoma ecotype categories in analyzed datasets according to *PTGS2*/*CTLA4* ratio expression (HIGH and LOW).

Ecotype Category	TCGA		AOCS		GSE26193		GSE0161	
	HIGH	LOW	HIGH	LOW	HIGH	LOW	HIGH	LOW
CE01 + CE02	45	52	15	9	12	6	9	2
CE03	4	4	-	1	1	1	-	1
CE04 + CE05	11	10	-	3	1	-	1	4
CE06	11	5	1	1	4	1	2	2
CE07 + CE08	38	27	11	8	7	11	8	6
CE09 + CE10	14	52	5	6	3	3	1	3
Total	123	150	32	28	28	22	21	18

Count (n) of Carcinoma ecotype (CE) categories for each analyzed dataset. HIGH and LOW ratio expression according to median expression (defined cutoff) of *PTGS2*/*CTLA4* ratio.

**Table 2 cancers-16-00080-t002:** Ecotyper NK cell states in analyzed datasets according to *PTGS2*/*CTLA4* ratio expression (HIGH and LOW).

NK Cell State	TCGA		AOCS		GSE26193		GSE0161	
	HIGH	LOW	HIGH	LOW	HIGH	LOW	HIGH	LOW
S01	16	38	6	3	3	6	2	-
S02	3	1	1	-	-	-	1	1
S03	34	36	9	9	7	4	3	1
S04	7	11	4	4	1	2	1	-
S05	14	11	3	-	2	1	2	1

Count (n) of NK cell states (S) for each analyzed dataset. HIGH and LOW ratio expression according to median expression (defined cutoff) of *PTGS2*/*CTLA4* ratio.

## Data Availability

Transcript per Million (TPM) RNA-seq expression and clinical data can be accessed and downloaded from Genomic Data Commons through the TCGA-biolinks R package and ICGC Data Portal for the AOCS cohort. Lastly, normalized RNA array data were downloaded from https://www.refine.bio/website (accessed on 20 July 2023).

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
