# Peer review of "Cyclooxygenase-2 Blockade Is Crucial to Restore Natural Killer Cell Activity before Anti-CTLA-4 Therapy against High-Grade Serous Ovarian Cancer"

_cancers, 2023, doi:10.3390/cancers16010080_

Round 1

Reviewer 1 Report (Previous Reviewer 3)

Comments and Suggestions for Authors

Actually, the authors added some experimental data regarding the NK cell cytotoxicity upon incubation with CBX or/and ipilimumab.  The NK cells have been isolated from two donors suffering from High-Grade Serous Ovarian Cancer (HGSOC).

The authors claimed that the NK cell activity was partly impaired by the action of PGE2 as inhibitors of PGE2 synthesis can increase the NK cell activity.

These additional data are of interest. Indeed, they suggest that blocking of PGE2 can influence the NK cell cytotoxicity.

Actually, the number of donors tested is limited and the relative low effect of ipilimumab can be related to the low expression or absence of CTLA-4 at the NK cell surface. Indeed, no experimental data regarding this expression are reported. Some information could be added as supplementary data.

Also, I understand the effort of the authors in trying of validating the bioinformatic and flow cytometry analyses. 

It would be valuable and significant to enlarge the number of patients analysed. Also, I would say that this point can be critical and instead of patients the authors could use NK cells from healthy donors. Indeed, it is conceivable that the blocking of PGE2 synthesis could be affected by the CBX.

Anyway, the authors should clearly indicate the limitations of the present analysis.

Comments on the Quality of English Language

English language is good.

Author Response

Dear Reviewer,

Please find our comments in the attached pdf. 

Kind regards,

Fernán Gómez-Valenzuela

Reviewer 2 Report (Previous Reviewer 2)

Comments and Suggestions for Authors

Thank you for going through the manuscript and the reviewers' points
In my honest opinion, the authors have responded satisfactorily to the reviewers’ criticisms.
The manuscript is well written and falls within the aim of this Journal.

Please correct reference 56 as follow:

Golia D'Augè T, Giannini A, Bogani G, Di Dio C, Laganà AS, Di Donato V, Salerno MG, Caserta D, Chiantera V, Vizza E, et al. Prevention, Screening, Treatment and Follow-Up of Gynecological Cancers: State of Art and Future Perspectives. Clin. Exp. Obstet. Gynecol. 2023, 50(8), 160. https://doi.org/10.31083/j.ceog5008160

Author Response

Dear Reviewer,

Please find our comments in the attached pdf. 

Kind regards,

Fernán Gómez-Valenzuela

This manuscript is a resubmission of an earlier submission. The following is a list of the peer review reports and author responses from that submission.

Round 1

Reviewer 1 Report

Comments and Suggestions for Authors

In this manuscript, authors conducted an integrated bioinfomatics analysis using four cohorts including 555 patients with HGSOC. They found that a high ratio of PTGS2/CTLA4 was significantly correlated with reduced survival of patients. Moreover, the increased COX-2 expressions were associated with NK dysfunction, and a low NK score was associated with poor survival of patients with non-responders and sub-optimal debulking. These results provide a hint for PTGS2/CTLA4 and NK cells as immunotherapeutic targets in HGSOC. However, there were several drawbacks in this study. In my opinion, some major revision should be resolved before accepted for publication.

1. The title of manuscript should be improved.

2. Please provide high resolution pictures in Figures, and the title of each figures in the upper of picture should be delete.

3. The numerical value in the figures and tables should be uniformly more than 3 decimal places.

4. Please explain GSE26197 n=62 (section of Abstract in Line 36), and KM survival curves for PFS in GSE30161? Actually, there were 107 samples in GSE26197, there was no PFS data in GDE30161.

5. The results from Figure 2 showed that a high ratio of PTGS2/CTLA4 was significantly correlated with reduced survival of patients. Please proved each KM plot of PTGS2 and CTLA4 in four cohorts in supplementary data.

6. Please explain why some datasets for cutoff were not median expression, such as GSE30161 of Figure 2, Figure 6F?

7. Please provide the graphic description for CE01-CE10 in Figure 3 and for S01-S05 in Figure 4.

8. please explain why PTGS2/CTLA4, not PTGS2*CTLA4, were further used to explore the correlation with patients with HGSOC. Because the transcriptional levels of PTGS2 were positively correlated with those of CTLA4.

Reviewer 2 Report

Comments and Suggestions for Authors

I read with great interest the Manuscript titled " Unraveling the COX-2/PGE2 Axis: Implications for Natural Killer Cell Activity and Anti-CTLA-4 Therapy Efficacy in High- 3 Grade Serous Ovarian Cancer - An Integrated Bioinformatics 4 Approach.”, topic interesting enough to attract readers' attention.

Although the manuscript can be considered already of good quality, I would suggest following recommendations: 

-       I suggest a round of language revision, in order to correct few typos and improve readability.

-       Considering topic analyzed, the authors could extend and improve the discussion by evaluating and citing current evidence about other possible target therapeutic strategies for patients with ovarian cancer.  I would be glad if the authors discuss this important point, referring to PMID: 37314974. 

-       Considering recent evidence in literature, it would be interesting to highlight the importance of prevention, screening and treatment of ovarian cancer with the aim to improve prognosis and the possibility of tailored management for these patients. I suggest authors to read and insert in references the following article: Tullio Golia D'Augè, Andrea Giannini,  Giorgio Bogani,  Camilla Di Dio,  Antonio Simone Laganà,  Violante Di Donato,  Maria Giovanna Salerno,  Donatella Caserta,  Vito Chiantera,  Enrico Vizza,  Ludovico Muzii,  Ottavia D’Oria. Prevention, Screening, Treatment and Follow-Up of Gynecological Cancers: State of Art and Future Perspectives. Clin. Exp. Obstet. Gynecol. 2023, 50(8), 160. https://doi.org/10.31083/j.ceog5008160

Because of these reasons, the article should be revised and completed. Considering all these points, I think it could be of interest to the readers and, in my opinion, it deserves the priority to be published after minor revisions.

Comments on the Quality of English Language

Minor editing of English language required

Reviewer 3 Report

Comments and Suggestions for Authors

This manuscript deals with the PTGS2/CTLA4 expression in ovarian carcinoma (specifically in high-grade serous ovarian cancer (HGSOC). The authors stated that the findings reported underscore the relevance of the COX-2/CTLA-4 axis in shaping the TIME and suggest its potential as both a prognostic indicator and therapeutic target. The findings reported are exclusively derived from a bioinformatic analysis of several cohorts of patients present in public databases. They actually need a validation, as indicated in the last sentence of the abstract.

Without this validation, this manuscript is not acceptable. It is well known the inhibitory effect of PTGS2 on immune reaction and its influence in shaping the tumor microenvironment (TME). That CTLA4 and the activity of COX2 are correlated is not surprising and not novel. The data shown can be performed well but actually are only descriptive, and they do not allow further interpretation.

 some points aslo are not clear at all:

For instance, what does it mean: "Notably, the S01 NK cell state was significantly lower in the PTGS2/CTLA4-high group in the TCGA ovarian cancer dataset"

" a lower NK state", what does it mean?

"We found that the total count of activated NK cells (estimated by MIXTURE)"

What is "Mixture" the authors should clarify on what it is based "mixture" Actually, I am an expert on human NK cells but anybody reading the paper should understand what Mixture is.

It is surprising that the focus of the manuscript would be NK cells, and no attempt to clarify what is what used for the analysis of this subset has been performed by the authors.

Comments on the Quality of English Language

English is good, at least in my opinion. But I am not an English language expert.

Round 2

Reviewer 3 Report

Comments and Suggestions for Authors

The authors replied to some of the concerns, but not the one regarding the validation of results. Certainly, they are going to demonstrate in the future with experimental data what they supported with the bioinformatic analysis. 

The data regarding the expression of Cox2 in NKG2A+ NK cells are of interest, as well as those on the CTLA4 expression. However, I do not consider the immunofluorescence data shown as exhaustive of the matter. 

I remain on my evaluation. This is a work well performed but limited.

Comments on the Quality of English Language

English is good, in general.